# Directed evolution of CRISPR-Cas9 to increase its specificity

Jungjoon K. Lee [1], Euihwan Jeong[2,3], Joonsun Lee[1], Minhee Jung[1], Eunji Shin[1], Young-hoon Kim[1], Kangin Lee[1], Inyoung Jung[1], Daesik Kim[3], Seokjoong Kim[1] & Jin-Soo Kim [2,3]

The use of CRISPR-Cas9 as a therapeutic reagent is hampered by its off-target effects. Although rationally designed *S. pyogenes* Cas9 (SpCas9) variants that display higher specificities than the wild-type SpCas9 protein are available, these attenuated Cas9 variants are often poorly efficient in human cells. Here, we develop a directed evolution approach in *E. coli* to obtain Sniper-Cas9, which shows high specificities without killing on-target activities in human cells. Unlike other engineered Cas9 variants, Sniper-Cas9 shows WT-level on-target activities with extended or truncated sgRNAs with further reduced off-target activities and works well in a preassembled ribonucleoprotein (RNP) format to allow DNA-free genome editing.

[1] Toolgen, Seoul 08501, Republic of Korea. [2] Center for Genome Engineering, Institute for Basic Science (IBS), Seoul 34121, Republic of Korea. [3] Department of Chemistry, Seoul National University, Seoul 08826, Republic of Korea. These authors contributed equally: Jungjoon K. Lee, Euihwan Jeong. Correspondence and requests for materials should be addressed to J.K.L. (email: jj.lee@toolgen.com) or to J.-S.K. (email: jskim01@snu.ac.kr)

The determination of the Cas9 crystal structure[1] enabled scientists to rationally design mutant Cas9 proteins (enhanced specificity Cas9 (eSpCas9)) and Cas9-high fidelity (Cas9-HF)) with higher specificities than wild-type Cas9 (WT-Cas9)[2,3]. Their design was based on the hypothesis that weakening non-specific interactions between a Cas9-RNA complex and its substrate DNA would reduce off-target activity. Since on-target activity is generally much higher than off-target activity, these mutant Cas9 variants would show higher specificities than WT while retaining on-target activity. However, it has been reported that both eSpCas9 and Cas9-HF are poorly active at some target sites[4–7], calling for alternative approaches to improve Cas9 specificity. More recently, two additional Cas9 variants, termed evoCas9 and HypaCas9, with improved specificity and activity were developed, reflecting unmet needs in this field[8,9]. In this study, we present Sniper screen, an *E. coli*-based selection method, to isolate SpCas9 variants with high specificity and activity and compare the resulting Sniper-Cas9 variant with other engineered SpCas9 variants.

## Results

**Simultaneous positive and negative selection using *E. coli*.** We reasoned that directed evolution of Cas9 in *E. coli* could lead to Cas9 variants with high specificity without killing on-target activities. The system we used consists of *E. coli* strain BW25141 and a plasmid containing the lethal *ccdB* gene[10–12] and the Cas9 target sequence: the disruption of the *ccdB* gene by Cas9-mediated plasmid DNA cleavage is essential for cell survival, creating a positive selection pressure. In addition, a Cas9 off-target sequence that differs from the on-target sequence by a few mismatches is introduced in the *E. coli* genomic DNA: double strand breaks (DSBs) in *E. coli* genomic DNA lead to cell death. We combined such negative selection pressure with *ccdB* plasmid-based positive selection pressure to develop "Sniper-screen," which selects for Cas9 variants with increased specificities. Note that Cas9 variants with poor on-target activities or poor specificities cannot survive in this selection system.

We first inserted a 500-bp PCR product containing an *EMX1* fragment into the genomic DNA of the BW25141 strain using a protocol involving transposase[13] (Fig. 1a and Supplementary Figure 1). The resulting BW25141-EMX1 strain was transformed with two plasmids (Fig. 1b): a plasmid-expressing *ccdB* under the control of the pBAD promoter, which is induced by arabinose, and a plasmid expressing an single-guide RNA (sgRNA) under the control of the pltetO1 promoter, which is induced by anhydrotetracycline (ATC). A target sequence with mismatches relative to the *EMX1* site was inserted into the *ccdB* plasmid and the matching guide sequence was inserted into the sgRNA plasmid. We chose these *EMX1* on-target and off-target sequences because off-target activities had been carefully tested at the *EMX1* site previously using a series of mismatched sgRNAs[14]. To screen for Cas9 variants without attenuated on-target activity, a 21-mer *EMX1* sgRNA with the 5′ guanine mismatched to the target sequence (gX20), which showed diminished on-target activities with engineered Cas9 variants, was used (Fig. 1b). In a Sniper screen, the resulting *E. coli* strain is electroporated with a pooled library expressing mutant Cas9 variants under the control of a CMV-pltetO1 dual promoter induced by ATC. There are four possible outcomes with respect to DNA cleavage in the *ccdB* plasmid and the genomic DNA (Fig. 1a): only a mutant variant of Cas9 that discriminates the on-target sequence present in the *ccdB* plasmid from the off-target sequence present in the *E. coli* genomic DNA can survive. Our system uses separate Cas9 and sgRNA plasmids, making it easy to change sgRNA-encoding and target sequence-containing

plasmids in subsequent rounds of selection. Because Cas9 can also be expressed in mammalian cells via the CMV promoter, it is possible to check the on-target and off-target activity of the pool obtained in each round. In addition, Cas9 and sgRNA expression are controlled by the pltetO1 promoter, allowing regulation of gene expression by up to 5000-fold[15]. By increasing the concentration of ATC, the screening conditions become more lenient for *ccdB* cleaving positive selection and harsher for genomic DNA cleaving negative selection. Such adjustments were necessary to find the optimum conditions at which control experiments using WT-Cas9 and a null vector showed a large window for cleavage for each target-sgRNA pair (Supplementary Figure 2).

**Construction of the Cas9 library and Sniper screen.** SpCas9 mutant libraries with random errors in the whole Cas9 sequence were constructed using three different kits, resulting in library complexities of up to 10^7 overall. Two independent sets of screenings were performed using the libraries (Supplementary Figure 3). The first set started with more lenient screening conditions and progressed toward more stringent conditions: DNA shuffling was performed in the middle of the process to enrich the diversity. The second set employed harsh conditions without DNA shuffling. After the final selection steps for both screening sets, the pooled libraries were tested in mammalian cells to measure the specificities of the Cas9 variants compared to WT-Cas9 (Supplementary Figure 4). The pooled libraries showed higher specificities than WT-Cas9; the first set was more specific than the other set. One hundred colonies were picked from both sets and the Cas9-encoding DNA sequences were fully sequenced. Three Cas9 variants were identified from the first set, which were designated clone-1, clone-2, and clone-3 (Supplementary Data 1). The mutations were dispersed throughout domains revealed in the crystal structure[1]; none of them overlapped with those in rationally designed Cas9 variants (Supplementary Figure 5). Site-directed mutagenesis analyses revealed that no single-point mutation drastically improved the specificity of WT-Cas9 (Supplementary Figure 6). Since we performed the DNA shuffling reaction in the middle of our screening, we assumed that the mutant with the best combination of point mutations survived in our screening.

**Clone selection and characterization.** Among the three different variants, clone-1 showed the highest on-target activity in human cells (Supplementary Figure 4). Because our major goal was to select Cas9 mutants without compromised on-target activity, we chose to characterize clone-1, which was named Sniper-Cas9. Sniper-Cas9, along with rationally designed Cas9 variants (eSp-Cas9 (1.1), Cas9-HF1, evo-Cas9, and Hypa-Cas9) and WT-Cas9, were tested in HEK293T cells at 12 target sites. Because the nucleotide at the 5′ terminus in sgRNAs, transcribed in vitro under the control of U6 or T7 promoters, is fixed to a guanine, the 5′ guanine could be either a match (G) or mismatch (g) to a target sequence. As shown previously[4–6,8,9], other Cas9 variants showed high activities only with GX19 sgRNAs, whereas Sniper-Cas9 maintained high on-target activities comparable to WT-Cas9 not only with GX19 or gX19 sgRNAs but also with truncated and extended sgRNAs (shown as gX20 sgRNA), regardless of a match or mismatch at the 5′ end (Fig. 2a, b, Supplementary Figures 7 and 8).

**Comparison of Sniper-Cas9 with other engineered Cas9s.** We next compared the specificities of Sniper-Cas9 with those of other Cas9 variants by measuring indel frequencies at on-target and off-target sites via targeted amplicon sequencing and calculating the

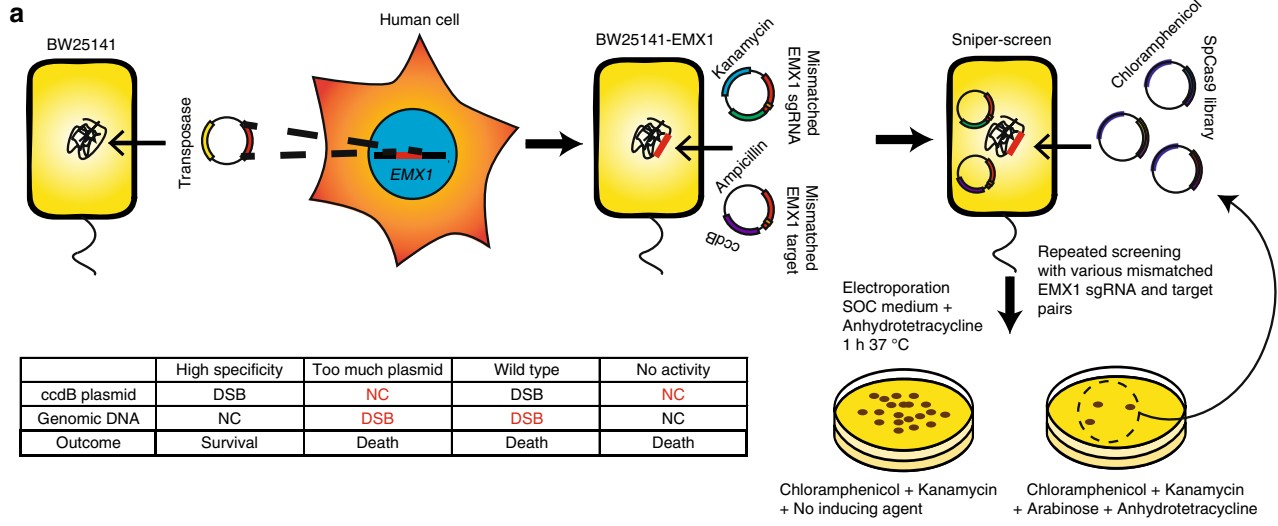

| | High specificity | Too much plasmid | Wild type | No activity |
|---|---|---|---|---|
| ccdB plasmid | DSB | NC | DSB | NC |
| Genomic DNA | NC | DSB | DSB | NC |
| Outcome | Survival | Death | Death | Death |

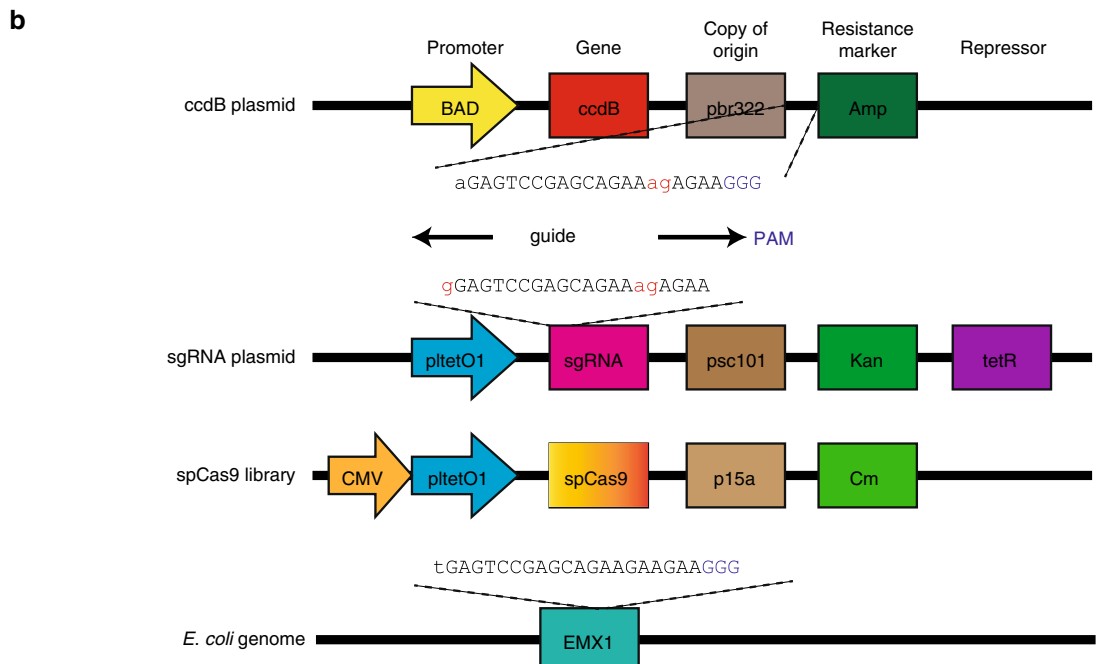

**Fig. 1** One-step positive and negative selection scheme in *E. coli*. **a** Schematic of the Sniper-screen. Cell death is indicated by red type. **b** Plasmid design with sample sgRNA and target sequences with mismatches in the fifth and sixth positions of the guide sequence targeting the EMX1 gene. DSB: double strand break, NC: no cleavage

ratios of on-target activity to off-target activity. In this analysis, we excluded Cas9 variant-sgRNA combinations with <70% on-target activity relative to WT-Cas9 in complex with 20-mer sgRNAs (Fig. 2c). Although we tested various combinations of Cas9 variants with truncated sgRNAs[16] or extended sgRNAs[17] to achieve the highest specificity, only Sniper-Cas9 and WT-Cas9 were compatible with these modified sgRNAs. Notably, Sniper-Cas9 achieved the highest specificity ratios in 10 out of the 12 on-target/off-target pairs. It should be noted that a specificity ratio higher than 1000 is above the detection limit of targeted deep sequencing using Illumina MiSeq (0.1%)[18]. In addition, the use of truncated sgRNAs is available for only a few targets because such sgRNAs worked at only 9 out of the 24 targets. To our knowledge, currently it is not possible to predict whether a truncated sgRNA will show any on-target activity or a higher specificity ratio prior

to its design. Western blot analysis showed that expression levels of WT-Cas9 and Cas9 variants were comparable in HEK293T cells (Supplementary Figure 9). We measured specificity ratios of Sniper-Cas9 and other Cas9 variants in Hela cells (Supplementary Figure 10). Similar results were obtained, showing that the high specificity of Sniper-Cas9 was not cell-line specific.

**Comparison of Sniper-Cas9 with xCas9–3.7.** More recently, Cas9 variants with broad PAM compatibility have been reported to show higher specificity[19]. We have tested 20-mer and 21-mer sgRNAs to characterize on-target and off-target activities of xCas9–3.7 (Supplementary Figure 12). xCas9–3.7 showed specificity ratios that were intermediate between those of WT-Cas9

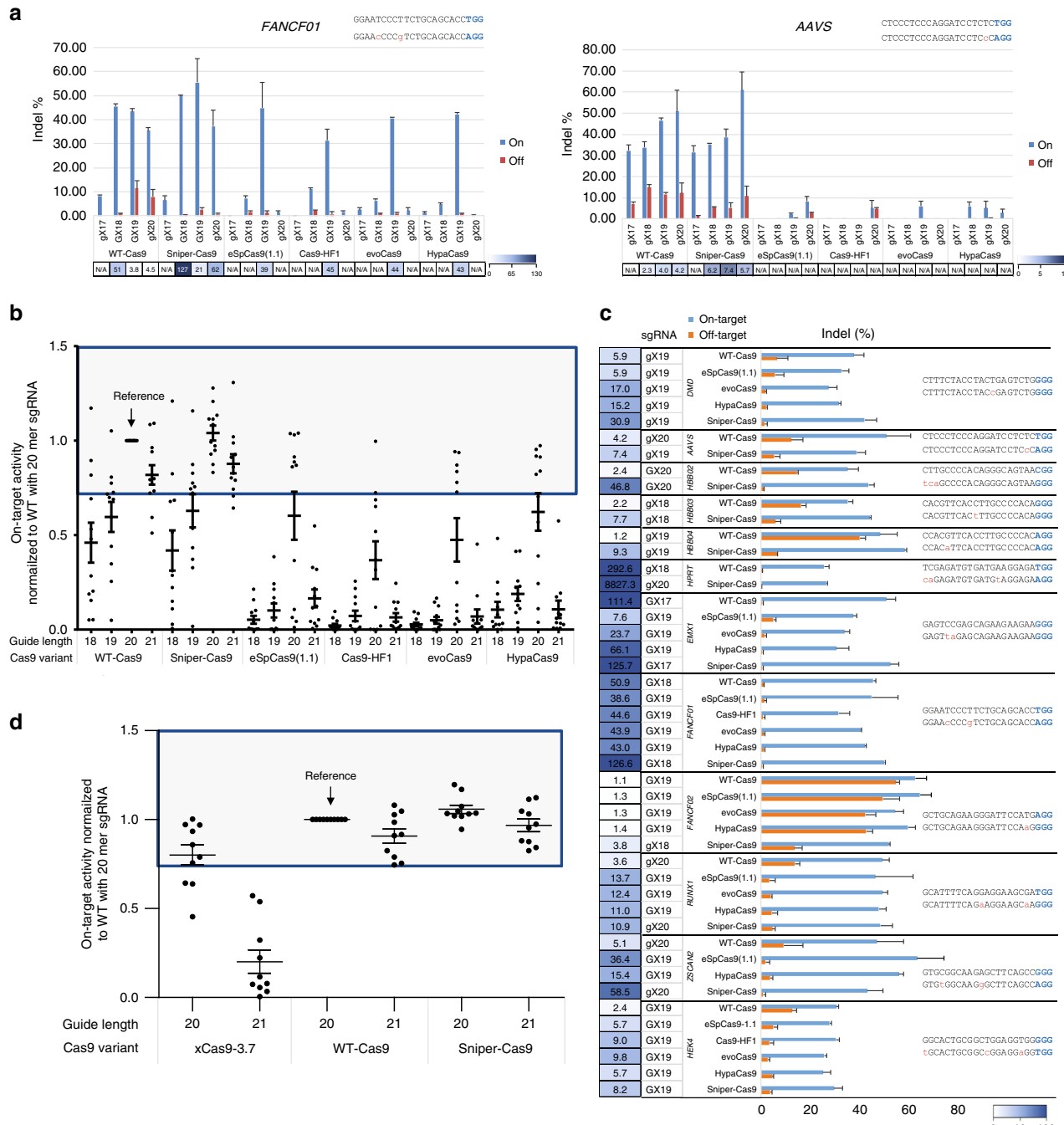

**Fig. 2** Sniper-Cas9 retains WT-level on-target activities with diminished off-target activities. **a** On-target and off-target activities of Cas9 variants compared to WT-Cas9 using sgRNAs with variable lengths targeting the *FANCF01* and *AAVS* sites. Specificity ratios were determined by dividing indel frequencies at on-target sites by those at the respective off-target sites. sgRNAs with a matched guanine at the 5′ terminus (GX18 or GX19) and those with a mismatched guanine (gX17, gX18, gX19, or gX20) are indicated. Specificity ratios were not calculated when the normalized on-target activities were <70%. **b** Dot plots of on-target indel frequencies normalized to those obtained with WT-Cas9 plus gX20 or GX20 sgRNAs. The gray-boxed area represents on-target activities higher than 70% of the WT-Cas9 activity. The numbers on the *x* axis represent the guide sequence length including the 5′ guanine; 18: gX17 or GX17, 19: gX18 or GX18, 20: gX19 or GX19, 21: gX20 or GX20. Error bars indicate s.e.m. (*n* = 12). **c** On-target and off-target activities of Cas9 variant-sgRNA combinations with >70% on-target activity relative to WT-Cas9. Specificity ratios were determined by dividing the on-target activity by the off-target activity. For each target, sgRNAs with the highest specificity ratios are shown for each Cas9 variant. Error bars indicate s.e.m. (*n* = 3). **d** Dot plots of on-target indel frequencies normalized to those obtained with WT-Cas9 plus gX20 or GX20 sgRNAs. The gray-boxed area represents on-target activities higher than 70% of the WT-Cas9 activity. The numbers on the *x* axis represent the guide sequence length including the 5′ guanine; 20: gX19 or GX19, 21: gX20 or GX20. Error bars indicate s.e.m. (*n* = 10). N/A: not available

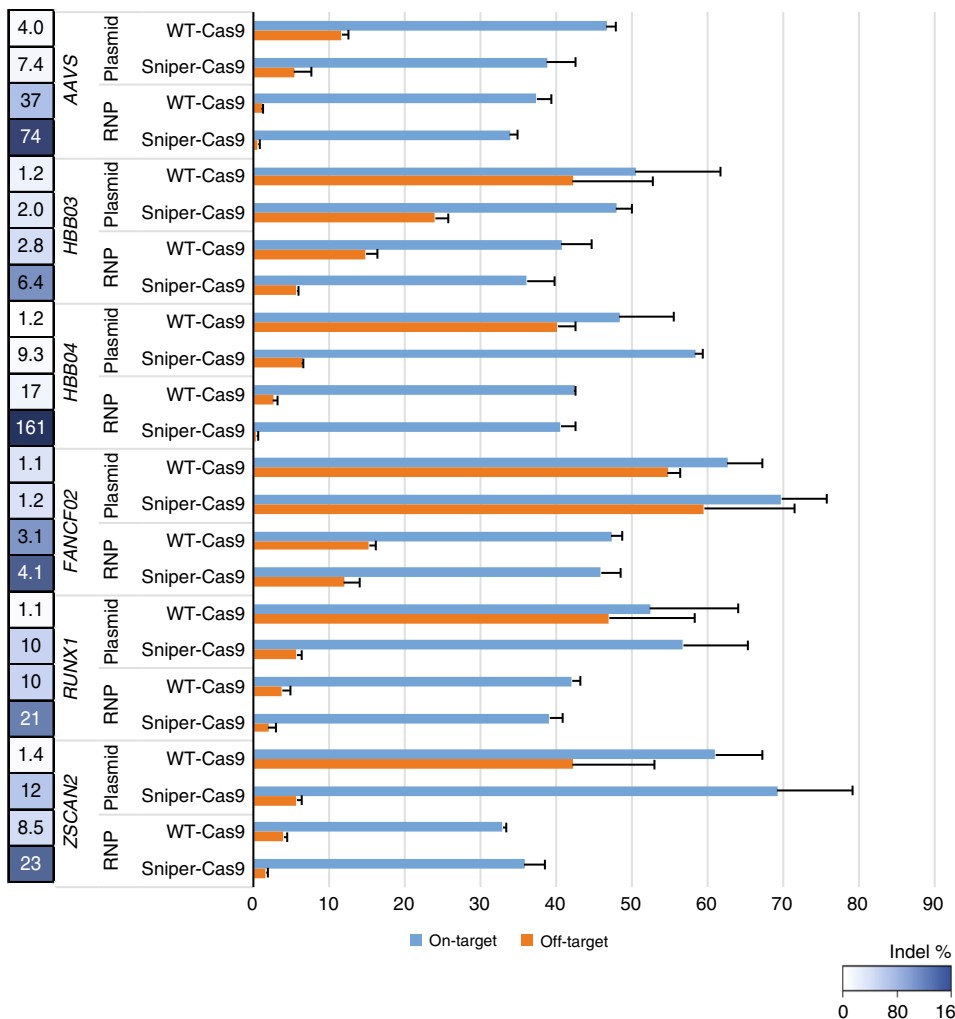

**Fig. 3** On-target and off-target activities of WT-Cas9 or Sniper-Cas9 paired with a 20-mer guide sequence delivered via plasmid or RNP. Specificity ratios were determined by dividing the on-target activity by the off-target activity. Error bars indicate s.e.m. ($n = 3$)

and Sniper-Cas9 at most targets and its on-target activities were attenuated significantly at many targets with mismatched 5′ guanines (Fig. 2d). It appears that although xCas9–3.7 exhibits a broadened PAM compatibility, from NGG to NG, it lost compatibility with a 5′ guanine. A recent mechanistic study revealed that the intrinsic cleavage rates of eSpCas9 (1.1) and Cas9-HF1 are 30 and 39 times slower, respectively, than the WT rate, which contributes to the higher specificity ratios of the engineered Cas9 variants[20]. It is speculated that the extensive mutagenesis of other engineered Cas9 variants including xCas9–3.7 lowers the intrinsic cleavage rate, resulting in lower on-target activity and a higher specificity ratio[21].

**On-target and off-target activities of Sniper-Cas9 RNP.** To further reduce off-target effects, we delivered Sniper-Cas9 into human cells in a preassembled RNP format[22]. We chose several of the on-target/off-target pairs with low specificity ratios shown in Fig. 2c that used 20-mer sgRNAs. Sniper-Cas9 RNPs were highly active and more specific than plasmids by a factor of 2–16 fold (Fig. 3).

**Mismatch tolerance of Sniper-Cas9.** We also tested a series of guide RNAs containing mismatches relative to the on-target sequence to investigate whether Sniper-Cas9 and other Cas9 variants tolerate single or double mismatches (Fig. 4a,

Supplementary Figure 11). Among three different targets tested, direct comparisons between all of the Cas9 variants and WT-Cas9 were possible only at the *FANCF01* target site owing to the low on-target activities of eSpCas9 (1.1) and Cas9-HF1 at other sites. For 13 out of 19 single mismatch positions, Sniper showed the highest specificity ratio. Sniper was outperformed by other engineered Cas9 variants in cases with mismatches at the PAM distal end (16th, 18th, and 19th). Almost no off-target activity was observed with Sniper-Cas9 combined with sgRNAs with double mismatches.

**Unbiased genome-wide off-target analysis of Sniper-Cas9.** Next, we performed multiplex Digenome-seq using four sgRNAs to test the genome-wide specificity of Sniper-Cas9 in the human gemome[23,24]. Sniper-Cas9 cleaved human genomic DNA at far fewer sites than did WT-Cas9 (Fig. 4b, c and Supplementary Figure 13). We analyzed off-target effects at candidate off-target sites that were uniquely cleaved by Sniper-Cas9 and that were cleaved by both Sniper-Cas9 and WT-Cas9 using targeted amplicon sequencing. No off-target mutations were detectably induced by Sniper-Cas9 at sites that were cleaved by Sniper-Cas9 alone (Supplementary Figure 15). Next, nine candidate sites, commonly cleaved by WT-Cas9 and Sniper-Cas9, with the highest DNA cleavage scores were selected for each sgRNA and off-target effects were validated at these sites by targeted deep

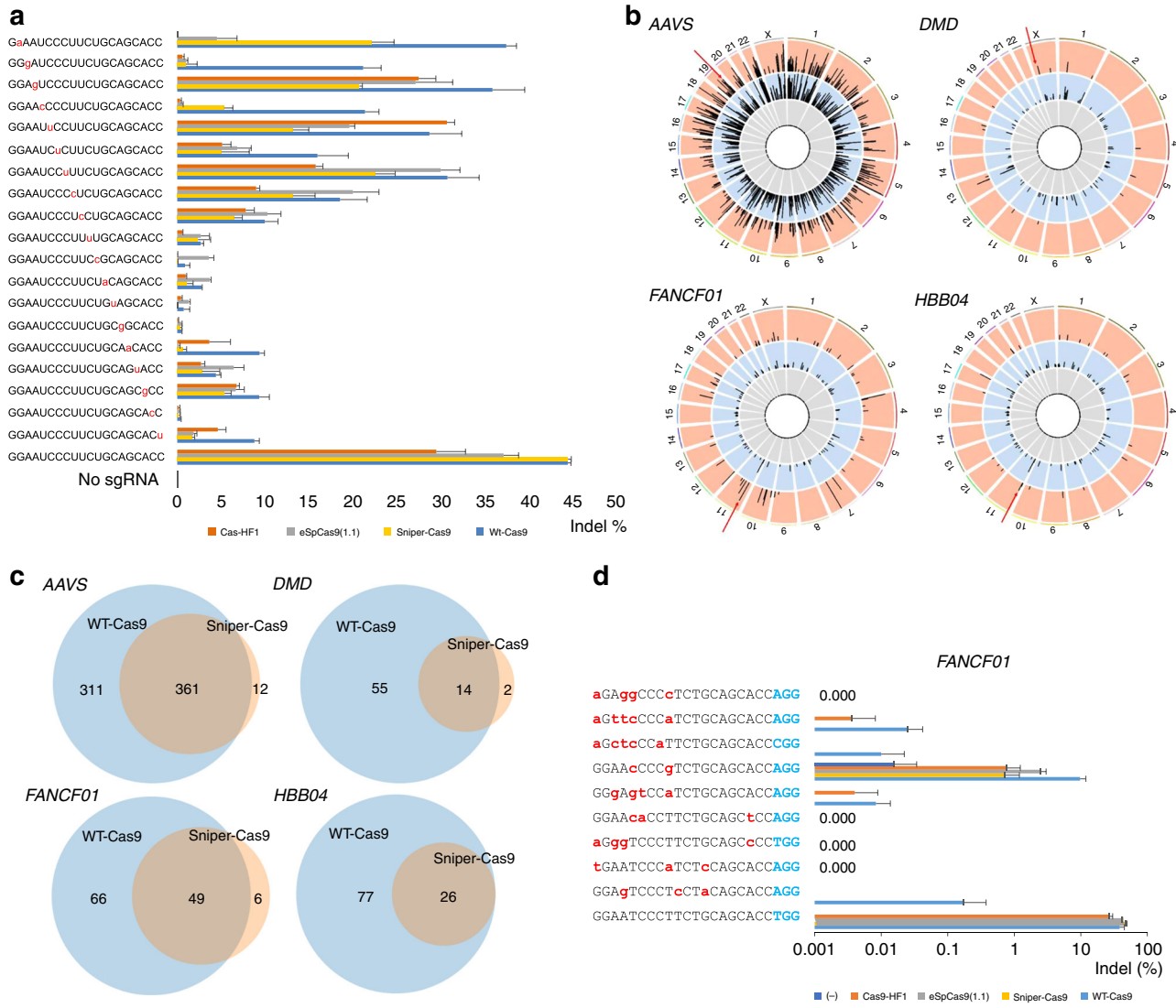

**Fig. 4** Unbiased genome-wide off-target analysis of Sniper-Cas9 using Digenome-Seq. **a** Tolerance of Cas9 variants for sgRNAs with mismatches relative to the *FANCF01* target. Frequencies of small insertions or deletions (indels) were measured using targeted deep sequencing. Error bars indicate s.e.m. ($n = 3$). **b** Genome-wide Circos plots representing DNA cleavage scores for *AAVS*, *DMD*, *FANCF01* and *HBB04* obtained with genomic DNA digested with untreated (gray), WT-Cas9 (blue), or Sniper-Cas9 (orange). Arrows indicate on-target sites. **c** Venn diagrams showing the number of in vitro cleavage sites captured by multiplex Digenome-sequencing analyses. **d** WT-Cas9, eSpCas9 (1.1), Cas9-HF, and Sniper-Cas9 off-target sites for *FANCF01* validated in HEK293T cells by targeted deep sequencing. Error bars indicate s.e.m. ($n = 3$) The PAM is shown in blue

sequencing (Fig. 4d, Supplementary Figure 14). Sniper-Cas9 showed lower than WT-level off-target activities at all sites that were analyzed. In conclusion, Sniper-Cas9 did not cleave any additional off-target sites in cells, compared to WT-Cas9.

Having confirmed the high genome-wide specificity of Sniper-Cas9, we further compared its specificity to that of other engineered Cas9 variants at additional GX19 targets. In this test, we measured off-target activities at validated targets from previous GUIDE-seq experiments (Supplementary Figure 16)[2,3,7,8]. Sniper-Cas9 showed WT-level on-target activities at all additional GX19 targets, whereas Cas9-HF1, HypaCas9, and evoCas9 showed <70% of the WT-level on-target activities at some targets. Sniper-Cas9 displayed a specificity comparable to that of other engineered Cas9 variants at most off-target loci and showed off-target activities that were under the detection limit. Sniper-Cas9 showed high off-target activities when the sgRNA mismatch was located at the PAM distal end. However, the mismatch tolerance at the PAM distal end also enables the use of truncated or extended sgRNAs with Sniper-Cas9 to achieve a higher specificity ratio at those sites,

as shown previously at the *HBB02*, *HPRT*, *EMX1*, *FANCF01*, and *ZSCAN2* targets (Fig. 2c).

**On-target and off-target activities of Sniper-Cas9 BE3**. We also investigated whether the mutations in Sniper-Cas9 can improve the specificity of base editors (BEs). To this end, Sniper-Cas9 mutations were introduced into BE3[25] to create Sniper-Cas9 BE3, which was tested in HEK293T cells to determine its base-editing efficiency (Fig. 5). Sniper-Cas9 BE3 was as efficient as WT-BE3 at the *EMX1* on-target site. At several pre-validated off-target sites that had been identified by Digenome-seq[26], Sniper-Cas9 BE3 showed much reduced off-target base-editing effects (2.4–16.2 fold less) compared to WT-BE3. In addition, the use of truncated sgRNA further eradicated off-target activities to near background level without significant loss of on-target activity. This result suggests that the nickase version of Sniper-Cas9 also exhibits higher specificity than the WT Cas9 nickase, without killing its on-target activity, unlike the nickase form derived from

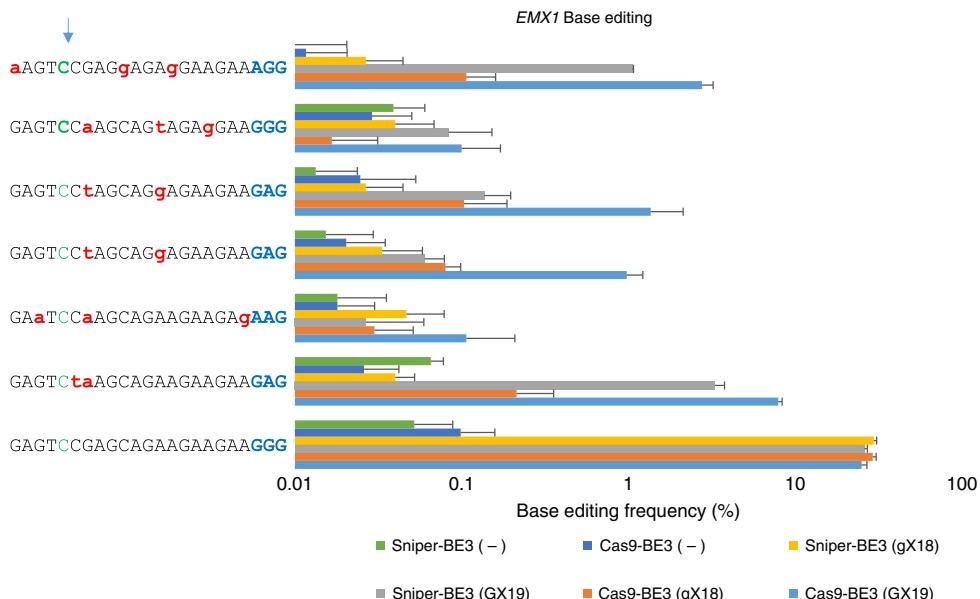

**Fig. 5** Base editor (BE3) on-target and off-target activities measured in HEK293T cells. (−) indicates the absence of sgRNA. Substitutions were measured using targeted deep sequencing. Substitution of C5 (represented by green type) to T was measured. The PAM is shown in blue. Error bars indicate s.e.m. ($n = 3$)

Cas9-HF (HF-BE3), which shows ~70% on-target activity compared to WT-BE3[27]. In addition, Sniper-BE3 showed WT-level activities with both GX19 and gX19 sgRNAs (Supplementary Figure 17), unlike BEs derived from Hypa-Cas9 and Cas9-HF1, which were reported to show low on-target activities with both forms[28]. Because the number of sgRNAs that could be used to edit specific bases in the target is limited, the use of Sniper-Cas9 would increase the number of targetable nucleotides compared to other engineered Cas9 variants by at least a factor of 4.

**Sniper-Cas9 RNP as potential therapeutic modality.** Finally, we tested the performance of Sniper-Cas9 RNPs as possible therapeutic agents. Primary human T cells and induced pluripotent stem cells (iPSC) were used in this proof-of-principle study. Although nucleofection of Cas9-encoding plasmid DNA into primary cells is possible, the low efficiency and high toxicity of this process represent a major hurdle for clinical applications[29,30]. The use of RNPs successfully resolved these issues in previous therapeutic development studies involving primary cells[31–34] with the additional advantage of reducing off-target activity compared to plasmid DNA[22].

The *AAVS* gene was selected as a hypothetical target because it is therapeutically relevant[35] and its use as safe-harbor site is not limited to a particular disease type. In addition, it is the most challenging target in terms of the number of candidate genome-wide off-targets identified by Digenome-seq (Fig. 4b) and the low specificity ratio of a validated off-target site containing a single mismatch (Fig. 2c). Finally, only Sniper-Cas9 showed WT-level on-target activity at this target locus, whereas all of the other engineered Cas9 variants failed (Fig. 2a).

We delivered into human T cells and iPS cells the purified Sniper-Cas9 protein in complex with 5′hydroxyl gX19 sgRNAs to further reduce toxicity caused by a triphosphate group present in in vitro-transcribed sgRNAs[36]. After RNP electroporation, no changes in cell morphology and the number of viable cells were observed, indicating that the RNP delivery was not cytotoxic (Supplementary Figure 20). Importantly, Sniper-Cas9 showed WT-level on-target activity with higher specificity ratios, compared to WT-Cas9 (1189 vs. 477), for the off-target site with

a single-nucleotide difference (Fig. 6). Both Sniper-Cas9 and WT-Cas9 did not induce off-target indels at any of the other top nine candidate sites found by Digenome-seq (Supplementary Figure 18), showing that off-target activities were cell-line dependent. Note that, gX19 sgRNAs were not compatible with other engineered Cas9 variants (Fig. 2a and Supplementary Figure 8) due to their lack of on-target activities caused by the mismatched 5′ guanine. In contrast, Sniper-Cas9 is compatible with 5′ mismatched or truncated or extended sgRNAs and can be delivered as purified, preassembled RNPs.

In summary, we developed Sniper screen in *E. coli* to create a SpCas9 variant with increased specificity and full on-target activity. It is anticipated that directed evolution of other Cas9 orthologues or Cpf1 by the Sniper screen would also generate highly efficient and specific derivatives ideal for therapeutic applications.

## Methods

**Plasmid construction.** Each type of plasmid used in the Sniper-screen contains replication origins and resistance markers that are compatible with each other. (Fig. 1b) The *ccdB* plasmid (p11-lacY-wtx1) was a kind gift from the Zhao lab[9]. It was double-digested with SphI and XhoI enzymes (Enzynomics), which was ligated to oligos (Cosmogenetech) containing target sequences (Supplementary Table 1) with T4 DNA ligase (Enzynomics). The sgRNA vector was constructed (Supplementary Figure 19) with a temperature-sensitive Psc101 replication origin[12] (from pgrg36, a kind gift from Nancy Craig), tetR (from the tn10 locus of ElectroTen-Blue Electroporation Competent Cells, Agilent), a Kanamycin resistance marker, the pltetO1 promoter and the sgRNA sequence containing two BsaI sites (synthesized at Bioneer). The components were PCR-amplified and Gibson assembled (NEBuilder HiFi DNA Assembly kit, NEB). The guide RNA sequences to *EMX1* with various mismatches (Supplementary Table 1) were cloned into the vector after BsaI digestion. The Cas9 library plasmid (Supplementary Figure 19) was derived from human codon-optimized WT-Cas9 (p3s-Cas9HC; Addgene plasmid #43945)[37], dual CMV-pltetO1 (synthesized at Bioneer) and the p15a replication origin and chloramphenicol resistance marker (from the PBLC backbone, Bioneer). The components were Gibson assembled.

**_EMX1_ genome insertion.** Human *EMX1* containing various sgRNA target sequences (~500 bp) was PCR-amplified and integrated into the pgrg36[12] vector between the NotI and XhoI sites (Supplementary Table 2). The cloned pgrg36-EMX1 vector was then transformed into the BW25141 strain to integrate the *EMX1* sequence into the tn7 site in the genomic DNA. EMX1-BW25141 was selected using the standard pgrg36 protocol[15] (Supplementary Figure 1).

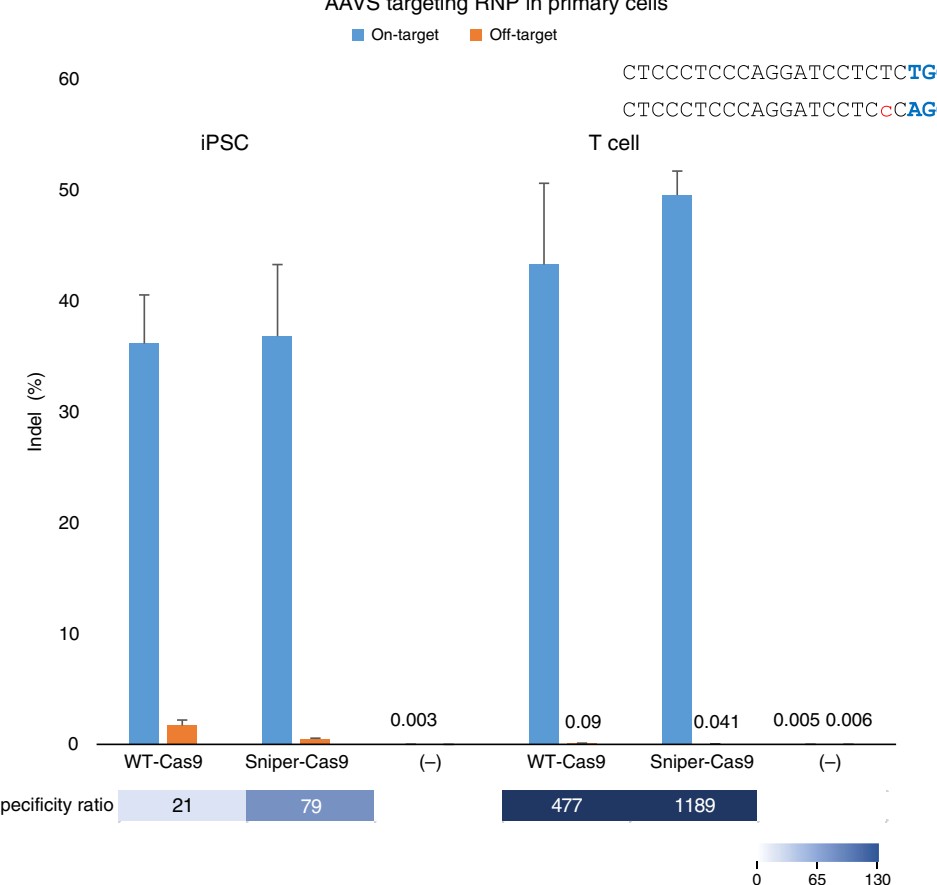

**Fig. 6** On-target and off-target activities of WT-Cas9 or Sniper-Cas9 paired with a gX19 guide RNA targeting *AAVS* and delivered via RNP into iPS cells and T cells. Specificity ratios were determined by dividing the on-target activity by the off-target activity. (−) indicates the absence of RNP. Error bars indicate s.e.m. (*n* = 3)

**Library construction**. SpCas9 mutant libraries were constructed using three independent protocols. For the first library, the Cas9 library plasmid was transformed into XL1-red competent cells (Agilent), which were grown according to instructions in the vendor's manual. For the second and third libraries, error-prone PCR was performed on whole WT-SpCas9 from Cas9 library plasmid sequences using Genemorph II (Agilent) and Diversify PCR random mutagenesis (Clontech) kits under low error rate (0–5 mutations per kb) conditions with primers designed for Gibson Assembly (Supplementary Table 3); PCR products were subsequently gel purified (4.3 kb). The purified randomly mutagenized library and the backbone of the Cas9 library plasmid (double-digested with BamHI and XbaI, followed by gel purification of the 3 kb fragment) were Gibson assembled. The assembled libraries were transformed into Endura™ electrocompetent cells (Lucigen) and incubated on chloramphenicol LB plates (12.5 μg/mL) at 37 °C overnight. A total of 3 × 10⁶ colonies were obtained for each library, resulting in a library complexity of 10⁷ overall. Pooled library plasmids were purified using a midi prep kit (NucleoBond Xtra Midi EF, Macherey-Nagel).

**Positive and negative screening for evolving SpCas9**. BW25141-EMX1 was co-transformed with the *ccdB* and sgRNA plasmids. The transformed BW25141-EMX1 cells were plated on ampicillin (50 μg/mL)/kanamycin (25 μg/mL) LB plates, which were then incubated overnight at 32 °C. Transformants were cultured in liquid S.O.B. medium containing 0.1% glucose, ampicillin, and kanamycin until the OD600 reached 0.4 for electrocompetent cell production (Sniper-Screen). Hundred nanograms of SpCas9 from each library was transformed into 50 μL of electrocompetent Sniper-Screen cells using a Gene Pulser (Gene Pulser II, Bio-Rad) following the manufacturer's instructions. Transformed Sniper-screen cells were mixed with 250 μL of S.O.C. medium. Twenty-five microliters of transformed cells was incubated without ATC (Sigma-Aldrich) and 250 μL of cells was incubated with 10 ng/mL ATC for 1 h at 37 °C. The Sniper-screen cells recovered in the absence of ATC were plated on chloramphenicol/kanamycin LB plates (non-selective conditions) and cells recovered in the presence of ATC were plated on chloramphenicol/kanamycin/arabinose (1.5 mg/mL, Sigma-Aldrich) LB plates (selective conditions) containing 100 ng/mL ATC followed by overnight culture at 32 °C. Viable colonies were counted using OpenCFU software[38] and the survival

frequency was calculated (survival frequency = the number of colonies on a selective plate/the number of colonies on a non-selective plate × 10). Colonies on the selective plates from three libraries were pooled and incubated in chloramphenicol-containing LB medium overnight at 42 °C to clear *ccdB* and sgRNA plasmids. Screened SpCas9 variant plasmids were purified using a midi prep kit (Macherey-Nagel) and 10 ng of pooled library was continuously transformed into the Sniper-screen until the survival frequency reached a plateau (Supplement Fig. 3). The ATC concentration in the selective conditions was maintained as 100 ng/mL for the two-mismatch conditions and as 10 ng/mL for the one-mismatch condition (Supplementary Figure 2). Selected SpCas9 gene variants obtained from the two-mismatch conditions were shuffled to increase library diversity (DNA-Shuffling Kit, Jena Bioscience) following the manufacturer's instructions. Screening of the shuffled SpCas9 library was performed again under the one-mismatch condition, and 100 colonies on a selective plate after six rounds of screening were individually cultured in chloramphenicol-containing LB medium at 42 °C to obtain evolved SpCas9 mutant plasmids. Each plasmid was Sanger-sequenced and the top three most frequent variants were chosen to be tested in a human cell line.

**Plasmids encoding Cas9 variants and sgRNA**. The WT-Cas9-encoding plasmid (p3s-Cas9HC; Addgene plasmid #43945)[37] and the sgRNA plasmid (pRG2; Addgene plasmid #104174)[4] have been described previously. Plasmids encoding human codon-optimized eSpCas9 (1.1) and Cas9-HF1 (p3s-eCas9 (1.1), Addgene plasmid #104172; p3s-Cas9-HF1, Addgene plasmid #104173)[4] were Gibson assembled into the p3s-Cas9HC plasmid backbone to change the location of the nuclear localization signal (NLS) from the N-terminus to the C-terminus. Human codon-optimized evoCas9, HypaCas9, and xCas9–3.7 (evoCas9, Addgene plasmid #107550, HypaCas9, Addgene plasmid #101178 and xCas9–3.7, Addgene plasmid #108379)[8,9,19] constructs were created by Gibson assembly of sequences containing the necessary site mutations into the p3s-Cas9HC plasmid backbone. All constructs were confirmed by Sanger sequencing (Supplementary Data 1). Human codon-optimized WT-BE3 and Sniper-BE3 were made by exchanging WT-SpCas9 from CMV-BE3 (a kind gift from David Liu; Addgene plasmid # 73021)[25] with WT-Cas9 and Sniper-Cas9 derived from p3s-Cas9HC.

**Cell culture and transfection conditions**. HEK293T cells (ATCC, CRL-11268) were maintained in DMEM medium supplemented with 10% FBS and 1% anti-biotics. For WT Cas9, eSpCas9(1.1), Cas9-HF1, and Sniper Cas9-mediated genome editing, HEK293T cells were seeded into 48-well plates at 70–80% confluency before transfection and transfected with Cas9 variants expression plasmids (250 ng) and crRNA plasmids (250 ng) using lipofectamine 2000 (Invitrogen). For base editing, HEK293T cells ($1.5 \times 10^5$) were seeded on 24-well plates and transfected at ~80% confluency with BE plasmid (Addgene plasmid #73021) (1.5 μg) or the Sniper-Cas9 expression plasmid and sgRNA plasmid (500 ng) using Lipofectamine 2000 (Invitrogen). Genomic DNA was isolated with the DNeasy Blood & Tissue Kit (Qiagen) 72 h post transfection.

**Recombinant Cas9 protein production**. Recombinant WT-Cas9 and Sniper-Cas9 proteins were purified from *E. coli*. The Cas9 DNA sequence was sub-cloned into pET28-b(+)[17]. Recombinant Cas9 protein containing a NLS, the HA epitope, and a His-tag at the N-terminus was expressed in strain BL21(DE3), purified using Ni-NTA agarose beads (Qiagen), and dialyzed against 20 mM HEPES pH 7.5, 150 mM KCl, 1 mM DTT, and 10% glycerol. The purified Cas9 protein was concentrated using an Ultracel 100 K cellulose column (Millipore). The purity and concentration of the Cas9 protein were analyzed by SDS-PAGE.

**Preparation of guide RNAs for RNP production**. RNA was in vitro-transcribed through run-off reactions with T7 RNA polymerase using a MEGAshortscript T7 kit (Ambion) according to the instructions in the manufacturer's manual. Templates for sgRNA or crRNA were generated by annealing and extension of two complementary oligonucleotides as described previously[52]. Briefly, sgRNA templates were generated by annealing two complementary oligonucleotides purchased from Macrogen. These oligonucleotides were reverse-phase-purified using the vendor's MOPC purification method and quality-checked using MALDI-TOF. sgRNA templates were incubated with T7 RNA polymerase in reaction buffer (40 mM Tris-HCl, 6 mM MgCl$_2$, 10 mM DTT, 10 mM NaCl, 2 mM spermidine, NTP, RNase inhibitor, at pH 7.9) for 8 h at 37 °C. Transcribed sgRNAs were preincubated with DNase I to remove template DNA, and purified using PCR purification kits (Macrogen). For 5′OH sgRNA generation only, the 5′-triphosphate was removed from guide RNAs with CIP (New England BioLabs) as follows: 10 μg of in vitro-transcribed RNA was treated with 250 units of CIP for 3 h at 37 °C in the presence of 100 units of RNase inhibitor (New England BioLabs). Following CIP treatment (or following in vitro transcription in the case of 5′PPP sgRNA), the RNA was cleaned up using a miRNeasy Mini kit (Qiagen).

**RNP delivery**. To introduce DSBs in HEK293T cells using an RNP complex, $2 \times 10^4$ cells were transfected with WT-Cas9 protein or Sniper-Cas9 (4 μg) premixed with in vitro-transcribed sgRNA (4 μg). To make RNP complexes, Cas9 protein in storage buffer (20 mM HEPES pH 7.5, 150 mM KCl, 1 mM DTT, and 10% glycerol) was mixed with sgRNA dissolved in nuclease-free water and incubated for 10 min at room temperature. RNP complexes were electroporated into HEK293T cells with a Neon transfection system (ThermoFisher) using the following settings: 1300 V, 30 ms, and 1 pulse. Genomic DNA was isolated with a DNeasy Blood & Tissue kit (Qiagen) 48 h post transfection.

**Western blotting**. The WT-Cas9, Sniper-Cas9, eSpCas9(1.1), Cas9-HF1, evoCas9, and HypaCas9 proteins expressed in HEK293T cells after transfection were detected using western blotting. Cas9 and GAPDH were detected using anti-HA (diluted 1:200, Santa Cruz Biotechnology, sc-7392) and anti-GAPDH (diluted 1:200, Santa Cruz Biotechnology, sc-32233) primary antibodies. Goat anti-mouse IgG-HRP antibody (diluted 1:1000, Santa Cruz Biotechnology, sc-2005) was used for signal detection. ImageQuant LAS4000 (GE healthcare) was used for digital imaging.

**In vitro cleavage of genomic DNA**. Genomic DNA was purified from HEK293T cells with a DNeasy Blood & Tissue Kit (Qiagen). Genomic DNA (10 μg) was incubated with Cas9 or Sniper1 protein (100 nM) and four sgRNAs (75 nM each) in a reaction volume of 1 mL for 8 h at 37 °C in a buffer (100 mM NaCl, 50 mM Tris-HCl, 10 mM MgCl$_2$, 100 μg/mL BSA, at pH 7.9). Digested genomic DNA was treated with RNase A (50 μg/mL) for 30 min to degrade sgRNAs and purified again with a DNeasy Blood & Tissue Kit (Qiagen).

**Whole-genome and digenome sequencing**. Genomic DNA (1 μg) was fragmented to the 400- to 500-bp range using the Covaris system (Life Technologies) and blunt-ended using End Repair Mix (Thermo Fischer). Fragmented DNA was ligated with adapters to produce libraries, which were then subjected to whole-genome sequencing (WGS) using a HiSeq X Ten Sequencer (Illumina) at Macrogen. WGS was performed at a sequencing depth of 30–40×. DNA cleavage sites were identified using Digenome 1.0 programs[18].

**Targeted deep sequencing**. Target sites and potential off-target sites (Supplementary Table 4) were analyzed by targeted deep-sequencing appropriate primers (Supplementary Table 5). Deep-sequencing libraries were generated by PCR.

TruSeq HT Dual Index primers were used to label each sample (Supplementary Table 6). Pooled libraries were subjected to paired-end sequencing using MiniSeq (Illumina).

**iPS cell genome editing**. iPSC generation has been described previously[39]. Briefly, BJ cells were cultured in DMEM supplemented with 10% FBS. A total of $1 \times 10^6$ BJ cells were resuspended in a single-cell suspension using trypsin and were electro-porated with three reprogramming plasmids (pCXLE-hOCT4-shp53 (Addgene plasmid #27077), pCXLE-hSK (Addgene plasmid #27078), and pCXLE-hUL (Addgene plasmid #27080)) using the Neon Transfection System (Invitrogen). Neon transfection conditions used were 1400 V, 20 ms, 2 time pulses for BJ cells. The transfected cells were seeded on a 6-cm dish and were cultured in BJ cell culture media for five days. The cells were then replated at a density of $1-3 \times 10^4$ cells per well in a 6-well plate pre-coated with vitronectin (STEMCELL Technologies) and cultured in BJ cell culture media for two more days; finally, the medium was changed to ips cell induction medium until all the iPSC colonies were harvested. CRISPR/Cas9-mediated genome editing of iPS cells (ATCC CRL-2522) was carried out as follows. iPSCs maintained on vitronectin-coated dishes (STEMCELL Technologies) in TeSR-E8 medium (STEMCELL Technologies) were detached using Gentle Cell Dissociation Reagent (STEMCELL Technologies). We used a four-dimensional nucleofector from Amaxa in combination with a P3 Primary Cell Kit for transfection. Four micrograms of recombinant *Streptococcus pyogenes* Cas9 (Toolgen) and 1 μg of 5′-OH sgRNA RNP were incubated for 20 min prior to electroporation to generate Cas9-gRNA RNP complexes. A total of $2 \times 10^5$ iPSCs re-suspended in P3 buffer were added to the pre-incubated Cas9-gRNA RNP complexes. Cells were nucleo-fected using program CA-137. Electroporation-only controls were nucleofected without RNP complexes using the same conditions. One microgram of enhanced green fluorescent protein (EGFP) messenger RNA (TriLink) was nucleofected into cells for the green fluorescent protein control under the same conditions. Cells were counted using Countess II Fl (Life technologies). Images of the cells were taken using an EVOS Fl Cell Imaging System (Thermo Fisher Scientific).

**T-cell genome editing**. CRISPR/Cas9-mediated genome editing of T cells was carried out as follows. Human peripheral blood pan-T cells were purchased from STEMCELL Technologies. Upon thawing, the T cells were allowed to rest overnight in RPMI supplemented with FBS, hrIL-2 (Peprotech, 50 U/mL), and hrIL-7 (Peprotech, 5 ng/mL) prior to activation. Activation was induced by the addition of Dynabeads Human T Activator anti-CD3/28 (ThermoFisher SCIENTIFIC) at a bead-to-cell ratio of 3:1 in RPMI supplemented with 10% FBS. 3 days later, the activating beads were removed and electroporation was carried out using an Amaxa P3 Primary Cell kit and 4D-Nucleofecter (Lonza). Eight micrograms of recombinant *S. pyogenes* Cas9 (Toolgen) and 2 μg of 5′-OH sgRNA were incubated for 20 min prior to electroporation to generate Cas9-gRNA RNP complexes. A total of $5 \times 10^5$ stimulated T cells re-suspended in P3 buffer were added to the pre-incubated Cas9-gRNA RNP complexes. Cells were nucleofected using program EO-115. Following electroporation, cells were seeded at $5 \times 10^5$ cells per mL in RPMI supplemented with 10% FBS, hIL-2 (Peprotech, 50 U/mL), and hIL-7 (Peprotech, 5 ng/mL). Electroporation-only controls were nucleofected without RNP complexes using the same conditions. Cells were counted using Countess II Fl (Life technologies). Images of the cells were taken using an EVOS Fl Cell Imaging System (Thermo Fisher Scientific).

**Data availability**. High-throughput sequencing data have been deposited in the NCBI Sequence Read Archive database SRR6374811.

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

## Acknowledgements

This research was supported by grants from the Institute for Basic Science (IBS-R021-D1) to J.-S.K., Ministry of Science and ICT of Korea (2017M3A9B4061406) and National Research Foundation of Korea (NRF) funded by the Korea government (MSIT) (grant nos. 2017M3A9B4061404 and 2018M3A9H3020844) to J.K.L. The plasmid encoding the pCMV-BE3 was a gift from David Liu (Addgene plasmid #73021), pGRG36 was a gift from Nancy Craig (Addgene plasmid #16666), and p11-LacY-wtx1 was a gift from Huimin Zhao. pCXLE-hOCT4-shp53 (Addgene plasmid #27077), pCXLE-hSK (Addgene plasmid #27078), and pCXLE-hUL (Addgene plasmid #27080) were kind gifts from Shinya Yamanaka.

## Author contributions

J.K.L designed and carried out directed evolution experiment (Sniper-screen) to obtain Sniper-Cas9. E.J., J.L., M.J., E.S., Y.h.-K., K.L., and I.J. characterized Sniper-Cas9. D.K. carried out bioinformatics analysis for Digenom-seq. J.-S.K., S.K., and J.K.L. supervised the research.

## Additional information

**Competing interests:** A patent application has been filed based on this work: Toolgen filed PCT/KR2017/006212 (Status: pending, Inventor: J.K.L.) covering Sniper-screen. J.-S.K. is a co-founder of and holds stock in ToolGen, Inc. J.K.L., J.L., M.J., E.S., Y.-h.K., K.L., I.J. and S.K. are employees of ToolGen, Inc. The remaining authors declare no competing interests.

