## [Peer Review File · Nature Communications]

Point-by-point Response

In response to reviewers' and editor's comments, we have now revised our manuscript. In this final revision, we made various textual changes to address various issues raised by the reviewer. Please find our point-by-point response below.

REVIEWERS' COMMENTS:

Reviewer #1 (Remarks to the Author):

Several approaches have been developed to decrease the off-target activity of SpCas9 such as using in vitro assembled RNP-s, shorter truncated gRNAs or mutant nuclease variants developed by rational design or directed evolution/selection. The manuscript "Directed evolution of CRISPR-Cas9 to increase its specificity" by Lee et al. describes an interesting approach similar to that published recently in Nature Biotechnology (ref), but using E.Coli instead of yeast. The authors developed an SpCas9 variant, sniper-cas9, that has increased fidelity in comparison to the WT but likely less attenuated than other increased fidelity nucleases such as e-, evo-, hypa- and -HF1 SpCas9s. They showed that sniper is compatible with 5' altered, extended or truncated gRNAs, considerably increasing the number of targets available for high fidelity nucleases. Furthermore, by combining sniper-Cas9 to other off-target-decreasing approaches, it outperforms the other high fidelity nucleases on several targets. One of the drawbacks of this approach, however, is that one needs to test several gRNA types for a target sequence to find out which one may have lower off-target effects. This likely will hinder its use. In addition, an SpCas9 variant (xCas9) is recently described (Hu et al., Nature vol 556 pg57 2018) with a more relaxed PAM that allows the use of an extended target space for the increased fidelity nucleases with even using G19XgRNAs. This decreases somewhat the significance of sniper. Nevertheless, sniper likely will become a useful complement of the other high fidelity nucleases for genome engineering tasks. I think, the paper could be suitable for publication in Nat. Com, in principle, provided the authors improve the MS on several points. Specifically, the authors do not seem to be satisfied by the performance of sniper and seem to over interpret/boost their data. In addition, there are too many mistakes/uncertainties in the data as presented and interpreted.

that somewhat makes difficult to have a clear assessment. The following list is an attempt to identify many of the issues in question but it is by no means comprehensive.

1. The authors state in the abstract that "iPS and T cells were successfully genome edited with Sniper-Cas9 with WT-level on-target activity and improved specificity." However, in reality, no off-target is detected in T cells using the WT protein, which makes difficult to improve specificity in this occasion. Since these are not the most impressive results of this study I suggest to leave it out from the abstract.

Response: We removed it from the abstract as suggested.

2. The authors do not seem to observe the detection limit of their experiments and use meaningless numbers to describe the fidelity of sniper. To avoid the false impression that the authors try to exaggerate the performance of sniper, I suggest to change the unrealistic numbers and the "infinite" specificities throughout the manuscript. This issue has been also raised in previous correspondences (ref2 further points 6.). For example, in line 209: "with high specificity ratios for the off-target site with a single nucleotide difference (Figure 6)" refers to the specificity ratio of 1189. In reality, the sniper on-target indel% is less than 50% at this target. Considering 0.1% detection limit for miseq, there is no meaning of a specificity higher than 500 (it should be indicated as >500).

Response: Detection limits are highly variable among target sites, ranging from 0.000% (Sup fig 18,) to 0.15% (sup fig 6, DMD mock) in this study alone. To determine detection limits, we always included a negative control (mock transfection). In Fig. 6, the detection limit was 0.006% (shown as (-)). As a result, the specificity ratios of WT-Cas9 and Sniper-Cas9 were 477 and 1189, respectively. To avoid confusion, we have now modified the sentence in question as follows: "Sniper-Cas9 showed WT-level on-target activity with higher specificity ratio, compared to WT-Cas9 (1,189 vs. 477) for the off-target site with a single nucleotide difference (Figure 6)."

3. It is not clear what off-target sequence is studied on Fig 4. It is specified neither in the text nor in the figure/figure legend. I would vote for the same sequence that is studied on Fig 2c using an educated guess. However, if this is the case, the authors should correct the sentence in line 210-212 stating the “remaining 10 targets” to “remaining 9 targets” since this target is included.

Response: This reviewer is right. We have now corrected it.

4. The authors claim in line 210 is misleading. “We did not detect indels at any of the remaining top ten (nine?) candidate off-target sites found by Digenome-seq, indicating that off-target activity occurred at frequencies under the detection limit (Supplementary figure 18).” The authors forget to mention that the WT SpCas9 showed also no off-target activity exceeding the detection limit (sup fig 18.)! These data should be properly interpreted!

Response: In response to this comment, we modified the sentence as follows: “Both Sniper-Cas9 and WT-Cas9 did not induce off-target indels at any of the other top nine candidate sites found by Digenome-seq (Supplementary Figure 18), showing that off-target activities were cell-line dependent.”

5. In line 154: “Next, ten candidate sites, commonly cleaved by WT-Cas9 and Sniper-Cas9, with the highest DNA cleavage scores were selected for each sgRNA and off-target effects were validated at these sites by targeted deep sequencing (Figure 4, Supplementary figure 14). ... In particular, no off-target indels were measurably induced by Sniper-Cas9 at Digenome-positive sites with two or more mismatches.” What is called measurable? On sup fig 15 and 18 less than 0.005 indel%-s are shown. These are much smaller than the detection limit. Here, there are sequences with 2 and even with 5 mismatches (HBBO4 (5.)) that reach about 1 indel%. That is above the detection limit. Are they not measurable? This needs to be reworded.

(Check sup fig 14; there are only nine, not ten off-target sequences for two out of the three on-target sequences.)

Response: Detection limits caused by sequencing errors are highly variable at each site. To determine detection limits, we included a negative control (shown as (-), mock transfection) at each site. “measurably-induced off-target indels” means indels induced above the limit at each site. We have now checked the number and corrected it.

6. The detection limit should be indicated by a solid line on all figures that show bars/values under the detection limit to avoid misinterpretations.

Response: Detection limits are now indicated in all figures as negative controls (shown as (-)).

7. Check sup fig 18.: 5 out of 11 and 7 out of 11 sequences have no indel% value/bar at all. At least noise level should be seen at this scale, at 0.001 indel%! Were these PCR products really present in the NGS sample?

Response: We have now indicated the numbers in case bars are too small. Please find attached three examples for 0 indel%, analyzed by Cas-Analyzer. Although large number of substitution errors are present in these samples, they are not included in indel %.

Cas-Analyzer

A JavaScript-based instant assessment tool for high-throughput sequencing data for genome edited cells.

Thanks to the improvements in the newest JavaScript engines in the most recent web browsers, the JavaScript based internal algorithm of Cas-Analyzer completely runs on the client-side so that large amounts of sequencing data do not need to be uploaded to the server. Currently, Cas-Analyzer supports various single nucleases (SpCas9, SsCas9, NmCas9, SaCas9, CjCas9, and AaCpf1/LbCpf1) and paired nucleases (ZFNs, TALENs, Cas9 nickases, and dCas9-FokI nucleases).

Citation info: Park, J. et al. Cas-Analyzer: an online tool for assessing genome editing results using NGS data. *Bioinformatics* 33, 286-289 (2017).

Input Summary

File1	File2
57_557_L001_R1_001.fastq.gz	57_557_L001_R2_001.fastq.gz
WT sequence (blue: indicator sequences at each ends of comparison range, green: crRNA sequence, red: WT marker sequence)	
ACAGGTGCATGCTCAAGTGTGACAGCTGCATAGGGGTGTATTCTGTCTCACTTCCAGTGGAGATCAAGGGATGGAAAGGAGTAGGGCCGAGAGGGGGTGTGTGTGGAGCCAGACCCAGGCTAGTGAAGTGTGCAGGCAGAGCCAGCCTGACCTTAGCACCCTTCCGACCTCTCCAGTGTACATCTTCTTGTGCCCTCAGTTATCCC	
crRNA sequence	
TCCCACCTGGATCCCTC	

Result Summary

Total Sequences	With both indicator sequences	More than minimum frequency	Insertions	Deletions	Indel frequency	HDR frequency
207544	11757	10566	0	0	0 (0.0%)	0 (0.0%)

WT and Substitutions Insertions Deletions Show HDR only

Download table

ID	Sequence	Length	Count	Type	HDR
1	ACAGGTGCATGCTCAAGTGTGACAGCTGCATAGGGGTGTATTCTGTCTCACTTCCAGTGGAGATCAAGGGATGGAAAGGAGTAGGGCCGAGAGGGGGTGTGTGTGGAGCCAGACCCAGGCTAGTGAAGTGTGCAGGCAGAGCCAGCCTGACCTTAGCACCCTTCCGACCTCTCCAGTGTACATCTTCTTGTGCCCTCAGTTATCCC	7325	WT or Sub	N/A	
2	ACAGGTGCATGCTCAAGTGTGACAGCTGCATAGGGGTGTATTCTGTCTCACTTCCAGTGGAGATCAAGGGATGGAAAGGAGTAGGGCCGAGAGGGGGTGTGTGTGGAGCCAGACCCAGGCTAGTGAAGTGTGCAGGCAGAGCCAGCCTGACCTTAGCACCCTTCCGACCTCTCCAGTGTACATCTTCTTGTGCCCTCAGTTATCCC	144	WT or Sub	N/A	
3	ACAGGTGCATGCTCAAGTGTGACAGCTGCATAGGGGTGTATTCTGTCTCACTTCCAGTGGAGATCAAGGGATGGAAAGGAGTAGGGCCGAGAGGGGGTGTGTGTGGAGCCAGACCCAGGCTAGTGAAGTGTGCAGGCAGAGCCAGCCTGACCTTAGCACCCTTCCGACCTCTCCAGTGTACATCTTCTTGTGCCCTCAGTTATCCC	48	WT or Sub	N/A	
4	ACAGGTGCATGCTCAAGTGTGACAGCTGCATAGGGGTGTATTCTGTCTCACTTCCAGTGGAGATCAAGGGATGGAAAGGAGTAGGGCCGAGAGGGGGTGTGTGTGGAGCCAGACCCAGGCTAGTGAAGTGTGCAGGCAGAGCCAGCCTGACCTTAGCACCCTTCCGACCTCTCCAGTGTACATCTTCTTGTGCCCTCAGTTATCCC	47	WT or Sub	N/A	

File1	File2
61_561_L001_R1_001.fastq.gz	61_561_L001_R2_001.fastq.gz
WT sequence (blue: indicator sequences at each ends of comparison range, green: crRNA sequence, red: WT marker sequence)	
CTTTCCCAAACTCCGATCAGGAGAAACCCACCCACCACTCCGCTCCATCCCAAGATCCCTCAGAACTCTTTTCCCGAAGAGACCTCGGAAAGGTGGCTCCCGCTCACTCTAGAAGTGAATCTCGACGCCCTCCCTCCGCGG9000	
crRNA sequence	
CACTTCCAGGATCCCTC	

Result Summary

Total Sequences	With both indicator sequences	More than minimum frequency	Insertions	Deletions	Indel frequency	HDR frequency
23312	13732	10442	0	0	0 (0.0%)	0 (0.0%)

ID	Sequence	Length	Count	Type	HDR
1	GAAGGAAACACCCACCACTCCGCTCCATCCCAAGATCCCTCAGAACTCTTTTCCCGAAGAGACCTCGGAAAGGTGGCTCCCGCTCACTCTAGAAGTGAATCTCGACGCCCTCCCTCCGCGG9000	4307	WT or Sub	N/A	
2	GAAGGAAACACCCACCACTCCGCTCCATCCCAAGATCCCTCAGAACTCTTTTCCCGAAGAGACCTCGGAAAGGTGGCTCCCGCTCACTCTAGAAGTGAATCTCGACGCCCTCCCTCCGCGG9000	1539	WT or Sub	N/A	
3	GAAGGAAACACCCACCACTCCGCTCCATCCCAAGATCCCTCAGAACTCTTTTCCCGAAGAGACCTCGGAAAGGTGGCTCCCGCTCACTCTAGAAGTGAATCTCGACGCCCTCCCTCCGCGG9000	110	WT or Sub	N/A	
4	GAAGGAAACACCCACCACTCCGCTCCATCCCAAGATCCCTCAGAACTCTTTTCCCGAAGAGACCTCGGAAAGGTGGCTCCCGCTCACTCTAGAAGTGAATCTCGACGCCCTCCCTCCGCGG9000	106	WT or Sub	N/A	

File1	File2
62_562_L001_R1_001.fastq.gz	62_562_L001_R2_001.fastq.gz
WT sequence (blue: indicator sequences at each ends of comparison range, green: crRNA sequence, red: WT marker sequence)	
TCTGCTCCTGTGGAGGATGATGTGATCCGACGGTCTCTGGAGGGCCATGACGGCTCCTCTGTCTGCTGAGCAGTGAAGGGTGCATCTTAGCATAGTTGGCAGCATCCCATTTGACCCACTCCCGCTCCTCTGACCTCCAGGACCTCCG	
crRNA sequence	
GACCTTCCAGGACCTCCTC	

Total Sequences	With both indicator sequences	More than minimum frequency	Insertions	Deletions	Indel frequency	HDR frequency
297823	21231	15653	0	0	0 (0.0%)	0 (0.0%)

ID	Sequence	Length	Count	Type	HDR
1	CATCTTAGCATAGTTTCCAGACCCATTGTGACCACTCCCGCTCTCTGTGACCTCCAGACCTCTCCGAGAGTGGGCTGATAGGTCATATTGTAAATTTGAGGCGGCAATGGCATCACTAGTGGTTC	8510	WT or Sub	N/A	
2	CATCTTAGCATAGTTTCCAGACCCATTGTGACCACTCCCGCTCTCTGTGACCTCCAGACCTCTCCGAGAGTGGGCTGATAGGTCATATTGTAAATTTGAGGCGGCAATGGCATCACTAGTGGTTC	292	WT or Sub	N/A	
3	CATCTTAGCATAGTTTCCAGACCCATTGTGACCACTCCCGCTCTCTGTGACCTCCAGACCTCTCCGAGAGTGGGCTGATAGGTCATATTGTAAATTTGAGGCGGCAATGGCATCACTAGTGGTTC	277	WT or Sub	N/A	
4	CATCTTAGCATAGTTTCCAGACCCATTGTGACCACTCCCGCTCTCTGTGACCTCCAGACCTCTCCGAGAGTGGGCTGATAGGTCATATTGTAAATTTGAGGCGGCAATGGCATCACTAGTGGTTC	205	WT or Sub	N/A	

8. The authors argue that sniper has high off-target activity with PAM distal and high fidelity with PAM

proximal mismatches, respectively, in comparison to other high fidelity nucleases. I am not convinced by the data and the authors' arguments on this. On fig3a, 3 out of the 6 positions at which sniper has lower activity compared to the other high fidelity nucleases fall between positions 17-19. I think their data is just not enough thorough to reach to such conclusion. A similar argument could be made for a lower PAM-distal specificity of SpCas9-HF relative to e-Cas9 based on the same data. e-Cas9 shows higher specificities than HF-SpCas9 at seven positions, among them four fall into positions 16-19. I do not think that it is a real difference.

Response: We have tested gX19 HBB02 and gX20 VEGFA in Supplementary figure 11 and found the similar tendency for Sniper-Cas9. (i.e. more off-target activities were observed in PAM distal mismatches compared to PAM proximal mismatches) In addition, test result of on-target activity result of gX18 targets (Figure 2a and Supplementary figure 8) is equivalent to 19-mer guide RNA test with mismatch at 19th position. As all the engineered Cas9 showed low on-target activities compared to WT-Cas9 unlike Sniper-Cas9s, we were further convinced that Sniper-Cas9 shows mismatch tolerance with PAM distal mismatches. It appears that this weakness is somewhat unavoidable result of our screening design. Sniper-screen was performed with gX20 guide RNA with mismatch at 5'end. If Sniper-Cas9 did not show mismatch tolerance with 5'g, it wouldn't have survived Sniper-screen due to low on-target activity resulting in low cleavage activity towards ccdB plasmid.

9. The authors made the experiments on fig 4 in response to the suggestion of Referee 1 to apply sniper to a real-world genome engineering problem and demonstrate success where other Cas9 variants fail. Transfecting cell populations and measuring indel% do not seem to be real-world genome engineering problems and the authors did not explain/prove why this job could not have been done by using other variant with an alternative gRNA.

Response: We admit that "a real-world problem" is a subjective phrase. We have now delete the phrase in the text. We also admit that the experiment we performed is not the most interesting example of real-world genome engineering that we removed this experiment from our abstract. However, we believe that there exists no target that interest every reader. Instead, the readers who do real world genome engineering might find this data relevant if they are worried about low on-target activities of other engineered Cas9 RNPs or conditions of the primary cells after electroporation step.

10. In response to Referee 2, the authors argue that monitoring the transfection efficiency is not necessary because the error bars over parallel samples show how stable was the transfection efficiency among samples. However, the transfection efficiency may vary very much among samples containing different DNA preps, what is not reported by the error bars. I suggest the authors to monitor their transfection efficiencies to make more trustable the performance of sniper.

Response: Thank you for your comments. We agree that monitoring transfection efficiency is important. As you mentioned, there are at least two sources of errors. One is the variation caused by slight difference in experimental conditions using the same material which we believe would be reflected in error bars. Second source of the variations is the material used in the experiment like cell line, lipofectamine2000 or plasmids that were used. When all the materials other than plasmids had problem, we observed low level of WT on-target activities compared to the previous values that we repeated our experiments with new materials. In this regard, we have actually monitored transfection efficiencies using WT on-target activities as the readout for positive control.

Furthermore, we have included Western blot results to show that WT and engineered Cas9 variants were expressed at equal levels in cells (Supplementary figure 9).